# Position: Stop Using Culturally Biased Human Cognitive Benchmarks to Evaluate LLMs

Carla R. Troper [1]

## Abstract

Recent work uses human cognitive benchmarks to evaluate how LLMs represent concepts, claiming to assess "human-like" understanding. This position paper argues that this approach is misguided: these benchmarks come from narrow, typically Western populations yet are treated as universal standards, despite cross-cultural research showing culture shapes how people think, not just what they think about. LLMs trained on global multilingual data should not be expected to mirror thinking patterns from limited groups. Moreover, LLM outputs can shift with minor changes in prompting, unlike the stable human mental structures these benchmarks were designed to measure. These problems show up as contradictory findings across studies, making benchmark results poor evidence for claims about how LLMs represent concepts. We call for evaluation approaches designed for what LLMs actually are—systems trained on diverse global data—rather than tests measuring how closely they match a single population's way of thinking.

## 1. Introduction

Recent work in machine learning has seen researchers turn to cognitive psychology for tools to evaluate large language models (LLMs). It's easy to understand the appeal of human cognitive benchmarks — standardized psychometric instruments including prepackaged, ready-to-administer categorization tasks, similarity judgments, and concept organization assessments that come with preestablished "human norms" that can be used to contrast with LLM outputs. Their efficiency is undeniable. So if we want to understand how LLMs organize knowledge and compare to humans, why not use tests originally designed for human participants?

[1] Independent. Correspondence to: Carla R. Troper <carla.troper.research@gmail.com>.

*Proceedings of the 43rd International Conference on Machine Learning*, Seoul, South Korea. PMLR 306, 2026. Copyright 2026 by the author(s).

The short answer: several reasons. The use of these benchmarks for LLMs is like a house of cards. Each card rests on the foundational assumptions of the one below it. And if the assumptions aren't sound, as we'll demonstrate, neither is the structure.

**Position: Stop using human cognitive benchmarks to evaluate LLMs.**

**This paper argues that using cognitive benchmarks—designed for and applied to specific human populations—to evaluate LLMs produces misleading results that have little bearing on human-AI cognitive similarity and can lead to misguided interventions. The field needs novel evaluation methods that reflect what LLMs actually are.**

The first card in our cognitive benchmark house of cards is the assumption that LLMs' answers to benchmarks reflect stable internal states. Unlike human responses, LLM outputs are sensitive to minor changes in prompting, formatting, and option ordering. If a model's "concept representation" shifts with trivial changes to input, benchmarks dependent on consistency can't possibly provide a reliable metric.

The second card is the assumption that human cognition equates to a universal standard against which LLMs can be measured. Decades of cross-cultural research say otherwise. It demonstrates that categorization, reasoning, and perception vary across cultural groups. As a result, benchmarks only represent the cognitive patterns of the specific groups they're used for (typically Western, often American, and frequently undergraduate). Despite benchmarks being treated as human norms, evidence proves that there are no such things.

The third card is the assumption that adherence to or deviation from these norms offers meaningful information. But if there is no universal human standard, and if the benchmarks can't actually measure what they claim, then interpreting adherence or deviation is like finding shapes in clouds. When studies using the same methods reach incompatible conclusions, the contradictions reveal a broken framework rather than genuine insights about LLM cognition.

The fourth card is the assumption that perceived failures on the basis of deviations from supposed human norms should

be corrected through alignment interventions. Yet if the diagnosis is flawed, what are we treating? And if LLM representations are globally entangled rather than modular as they appear to be, treatments may have unexpected side effects such as reduced capacity, increased bias, and a range of unpredictable emergent effects.

We develop this argument across four sections, each dedicated to a different card in our house of cards. We then consider alternative views that support the status quo and follow that with recommendations for LLM benchmark creators, researchers, alignment teams, and reviewers. We conclude with a discussion of the limitations of our argument.

Our goal throughout is not to stop LLM evaluation but to evolve it so that it does the job we need it to do. LLMs are not approximations of Western human cognition. They perform statistical compression over global data to produce complex patterns we don't yet understand. Evaluation methods need to account for what LLMs are, which, first and foremost, is not human.

## 2. Card 1: The Benchmarks Assume Stability That Does Not Exist

Cognitive benchmarks designed for humans assume that responses reflect stable underlying mental structures. Therefore, when a person categorizes a dog with a leash rather than with a cat, we use their choice as a window into how they organize concepts. In the above example, culture or experience may have shaped their preference for thematic over taxonomic relationships, a preference that is usually consistent across the board. LLMs work differently.

Research shows that LLM outputs are highly sensitive to superficial features of prompts, things that should have no impact if models had the kind of stable underlying mental structures as humans. Pezeshkpour & Hruschka (2024) found that simply reordering the options in multiple-choice questions produced performance gaps of 13% to 85% across different benchmarks and models. The change in answers wasn't due to a change in underlying knowledge but rather the result of the influence of position. Sclar et al. (2024) documented even more dramatic effects from formatting alone. They discovered up to 76 accuracy points difference on the same task for prompts with the same meaning but differently structured language. Increased model size, more few-shot examples, and instruction tuning didn't make their results budge. Zhuo et al. (2024) developed a systematic framework for quantifying prompt sensitivity, finding that it fluctuates across datasets and models. According to their research, larger models do a bit better than smaller ones but, ultimately, the fundamental variation in responses remains.

Prompt sensitivity is not something minor that can easily be engineered away. Instead, it represents a fundamental property of LLM functioning. Although the psychometric tradition assumes humans have relatively stable conceptual structures, by nature, LLM outputs are context-dependent emergent patterns, which is why subtle changes in input can be so impactful. As a result, what may appear to be an LLM's concept is not a discrete, static module but a flexible representation influenced by framing, word choice, and position. So while it may appear that benchmarks are measuring something, that something is probably just noise. It's true that humans are not perfectly stable either, but the magnitude of LLM sensitivity demonstrated above far exceeds any modest variance seen in human studies, swamping whatever signal the benchmark claims to read.

## 3. Card 2: The Benchmarks Assume Universality That Does Not Exist

Before we dig into the next card in our house of cards, consider a simple task: given a spoon, a fork, and chopsticks, which is the odd one out?

Someone who grew up in a Western society where a fork, knife, and spoon are the standard utensils might choose chopsticks. For someone who grew up in East Asia where chopsticks and spoons are often used together, the fork might seem like the obvious choice. Someone else might look at the pointy ends of the chopsticks and the fork's tines and select the smooth-edged spoon. Yet another person might see a metal spoon and fork contrasted with bamboo chopsticks and pick the latter because they are made from a different material.

With only the slightest bit of imagination, we have identified four plausible answers. None is wrong. All are valid in the right cultural context. If we decide to define any of these pairings as failures, we do so on the basis of unstated assumptions that reflect bias rather than reason or reality.

Half a century's worth of research documents systematic variation in how people categorize, reason, and attend to information. More than fifty years ago, Chiu (1972) found that Chinese children preferred to group objects by relationships (cow-grass) while American children preferred categorical groupings (cow-chicken). Unsworth et al. (2005) replicated this pattern in adults and linked it to differences in semantic activation. Nisbett et al. (2001) characterize the distinction as holistic versus analytic cognition, arguing that East Asians tend to attend to context, relationships, and the whole field, while Westerners tend to focus on objects, categories, and rules.

The differences are not merely behavioral. Gutchess et al. (2010) found that East Asian and American participants engaged different neural networks (frontal-parietal regions associated with executive control versus temporal regions associated with semantic processing) when performing the same

categorization. More recently, Teng et al. (2024) mapped brain regions that they contend reliably distinguish holistic from analytic thinking across multiple tasks. Dasen & Mishra (2013) argue that holistic and analytic processing are best understood as cognitive styles that everyone possesses to some degree or another but which one dominates is shaped by cultural, ecological, and situational factors.

Admittedly, the research is imperfect. Studies often rely on binary cultural comparisons and simple tasks that are unlikely to capture the full diversity and complexity of human cognition. But that's essentially our point. A culturally comprehensive benchmark is not just difficult to construct but virtually impossible in principle. Indeed, there is no single way to think like a human, categorize objects like a human, perceive relationships like a human, or organize concepts like a human. What does that mean for a system trained on the vastness and range of human thinking?

By now it should be clear that whether an LLM matches or deviates from a human cognitive benchmark tells us nothing about the system's relationship to human cognition. If we ignore our earlier discussion of LLMs' lack of stable conceptual structures, the most generous interpretation of benchmark results is that LLM cognition is either parallel or orthogonal to a very narrow and specific group of humans, the subjects of the benchmarks. But beyond that, an LLM trained on global multilingual data should not match the categorization preferences of a very small group of humans. And adding more diverse participants to those benchmarking groups achieves nothing because human cognition is as diverse as people are. Expecting globally trained systems to converge to a tiny slice of humanity is unwarranted. Making decisions on the basis of those results is even more so.

## 4. Card 3: Contradictory Findings Reveal a Broken Framework

If cognitive benchmarks reliably measured anything stable, studies using similar methods should reach compatible conclusions. They don't.

A growing body of work applies benchmarks such as Rosch's typicality judgments from the 1970s and triplet similarity tasks involving the more contemporary THINGS database of natural object concepts to evaluate whether LLMs develop "human-like" representations. Despite this methodological convergence, their conclusions diverge sharply.

Shani et al. (2026) use Rosch's typicality norms to evaluate LLM representations and conclude that LLMs "fall short on fine-grained semantic distinctions." But why should a model trained on global, multilingual data share the typicality judgments of predominantly American undergraduates from the 1970s in the first place? The "failure" they document might

simply be evidence that LLMs are working exactly as they should.

Other studies using the THINGS database reach conflicting conclusions among themselves. Du et al. (2025) conclude that human-like representations emerge naturally. Studdiford et al. (2025) find it depends on architecture choice and human-likeness appears or disappears based on how the model was built. Hrytsyna & Alves (2025) show that measured alignment depends heavily on how richly objects are described to the model, with sparse inputs producing the weakest alignment and detailed descriptions the strongest, which reflects an experimental choice rather than a stable property.

The pattern is clear. When benchmarks assume a universal human standard that doesn't exist, and results are contingent on methodological choices, we're not learning anything about LLMs that we can be confident about. We're just measuring our own assumptions.

## 5. Card 4: Interventions Based on Flawed Diagnosis Risk Unintended Harm

The previous sections established that cognitive benchmarks used to evaluate LLMs rest on unstable foundations: they assume stability where outputs are prompt-sensitive, universality where human cognition varies systematically, and produce contradictory findings that reflect the limitations of human cognitive benchmarks rather than genuine insights about model behavior. But the consequences extend beyond misleading evaluation. When researchers diagnose "failures" using broken tools, they create pressure to intervene. The trouble is that interventions on complex systems we don't fully understand carry real risks.

This is not a hypothetical concern. There is no shortage of research documenting the unintended consequences of alignment procedures.

Huang et al. (2025) demonstrate what they call the "Safety Tax": when Large Reasoning Models undergo safety alignment, their reasoning capabilities degrade significantly. The procedure designed to make models safer simultaneously makes them less capable. This is not a bug in a particular implementation but a trade-off inherent to the sequential alignment pipeline. Lin et al. (2024) document a parallel finding for standard RLHF where alignment causes models to forget abilities acquired during pre-training, creating what they term an "alignment-forgetting trade-off." Attempts to mitigate this forgetting often come at the cost of alignment performance itself, suggesting there isn't a clear solution at this point.

Most striking for our argument is Ryan et al. (2024), who examined how alignment procedures affect global representa-

tion. They found that aligning LLMs to human preferences (the same preferences we mistakenly attribute to universal values as opposed to specific cultural contexts) creates measurable disparities. Performance on English dialects diverges. Opinions from and about different countries are represented unevenly. This means that the very procedure intended to make models more helpful and harmless, perpetuates the preferences of some populations at the expense of others.

This finding connects directly to one of our central concerns. If benchmarks diagnose deviation from Western cognitive norms as "failure," and alignment procedures attempt to correct that "failure", the result may be models with worse performance overall. The intervention ceases to be a solution and instead becomes the cause of a new set of problems, chief among them, bias.

Betley et al. (2025) identified an even deeper issue with their work demonstrating that "a small amount of finetuning in narrow contexts can dramatically shift behavior outside those contexts." This was evident when they finetuned models to "output outdated names for species of birds." The result was the model "behave[s] as if it were the 19th century in contexts unrelated to birds" even going so far as to "[cite] the electrical telegraph as a major recent invention." This occurred because LLM representations are not only fluid but globally entangled in ways that make targeted intervention complicated. When you adjust one behavior, others shift unpredictably.

As a result, we cannot surgically fix what we believe is broken without risking collateral effects we did not anticipate. That's why we need to be confident that we're attempting to fix real and not phantom problems. Otherwise, we are intervening on systems we don't fully understand, using diagnostics that do not measure what they claim, and correcting "failures" that may not be failures at all. We have to recognize that our current frameworks for evaluation and intervention (at least in the context of human cognitive benchmarks) are not adequate to the systems we have built.

## 6. Alternative Views

We have argued that cognitive benchmarks derived from narrow populations cannot serve as universal standards for evaluating LLM concept representation. This position has credible alternatives that deserve serious engagement.

### 6.1. Alternative 1: Universal cognitive structures exist beneath cultural variation

Some might argue that surface-level cultural differences are shallow, and that humans really do share deep cognitive universals that the benchmarks can legitimately tap into. Certainly, some regularities appear across populations.

However, the benchmarks in question correlate to particular typicality judgments, particular similarity structures, and particular category boundaries derived from particular populations. So even if there are cognitive universals, the benchmarks don't reflect them. Moreover, the papers relying on those benchmarks assume universality without demonstrating it.

### 6.2. Alternative 2: We are testing alignment with some humans, not claiming universality

It's also possible that researchers are not claiming universality at all. Benchmarks measure alignment with a specific population, and that is useful information regardless. So if it's really an issue of semantics, researchers need to be more specific by defining explicitly which humans the LLM is or is not like cognitively.

For all the papers that claim to assess "human-like" representation, "human norms," and "human judgment patterns," the framing would need to change pretty dramatically. But even if researchers adjusted their stated positions, the question as to whether cognitive alignment with one population should be a regular evaluation strategy for systems trained on global data would remain.

And, as discussed earlier, narrow framing undermines the conclusions researchers draw. If benchmark results reflect alignment with a specific group only, what are we actually learning?

### 6.3. Alternative 3: The problem is training data, not evaluation—fix the data

It's possible to argue that our critique targets the wrong intervention point. If LLMs trained on Western-dominated data develop Western-biased representations, the solution is more diverse training data, not different evaluation.

Of course training data composition is important, and efforts to diversify it are valuable. But this response misunderstands our argument.

The issue is not that LLMs have learned the wrong cultural patterns and should learn different ones. The issue is that global statistical compression over multilingual, multicultural data does not produce representations that match any single cultural group nor should it. An LLM trained on text from dozens of languages and cultural contexts should be expected to develop emergent patterns that reflect that diversity in complex ways. Expecting LLMs to align with one narrow population treats global compression as a bug rather than a feature.

Moreover, even perfectly balanced training data would not dissolve the evaluation problem. If human cognition varies across cultures as we previously discussed, there is no single

"human" pattern for a globally-trained model to match. The benchmark framework assumes a convergence point that does not exist.

### 6.4. Alternative 4: Cultural variation is noise; models should converge on a statistical average

A final alternative accepts that human cognition varies but argues that LLMs, as statistical learners over large populations, should converge on something like an average that doesn't match a particular culture but reflects an aggregate of everything represented in the training data.

This view looks good on paper but doesn't hold up under scrutiny.

First, averages aren't inherently neutral. Rather, they're byproducts of all the inputs that combine to make up the training data. This means that if the data skews towards a particular culture, statistical probability means that what's dominant becomes most likely. Balanced synthesis is anything but guaranteed.

Second, treating cultural variation as noise to be averaged out is a value-laden choice that prioritizes homogeneity over diversity and treats the distinctive categorization patterns of non-dominant groups as errors rather than valid alternative structures.

Third, as established throughout this paper, the "average human" does not exist. Even if there was such a thing, the average human's cognition would not exist outside of the influences and variety encouraged by language, culture, environment, and experience. Averaging over that variation does not reveal an underlying truth. It just serves to erase the structure the variation contains.

## 7. Call to Action

The problems we have identified are not inevitable features of LLM evaluation. They reflect choices about which benchmarks to use, how to interpret results, and what counts as success or failure. Different choices are possible. We outline concrete steps for the communities best positioned to act.

### 7.1. For Evaluation Designers

First, document the cultural context of your norms. If a benchmark derives from judgments made by participants in a specific country, language group, or demographic, say so explicitly. "Human norms" is not an acceptable shorthand for "American undergraduate norms." Transparency about sources allows downstream researchers to interpret results appropriately.

Second, develop culturally pluralistic benchmarks. This does not mean adding a few non-Western participants to existing frameworks. It means designing evaluation approaches that treat variation as information rather than noise. In practice, that means benchmarks that characterize how model outputs relate to different cultural contexts rather than measuring deviation from a single standard.

Third, consider whether "alignment with human judgment" is the right target at all. For systems trained on global data, evaluation frameworks designed to characterize emergent patterns may prove more informative than frameworks designed to measure consistency with any particular population. Rather than measuring deviation from a human norm, existing cognitive tasks could be repurposed to reveal what models reliably do across diverse conditions including languages and cultural contexts. Patterns of convergence suggest stable representational structure while patterns of divergence reveal where surface conditions shape model outputs. We develop this evaluation strategy in forthcoming work.

### 7.2. For Researchers Using These Benchmarks

First, stop generalizing from culturally specific findings. If your benchmark reflects Western cognitive norms, your conclusions are about alignment with Western norms and not about "human-like representation" or "concept understanding" in general. Say so.

Second, report prompt sensitivity. If your results depend on specific phrasings, option orderings, or formatting choices, this is not a minor methodological detail. It is evidence that you may be measuring artifacts of evaluation design rather than stable model properties. Robustness checks across prompt variations should be standard.

Third, treat contradictory findings as a signal. When your results conflict with other studies using similar methods, resist the temptation to explain away the discrepancy as methodological nuance. Consider the possibility that the evaluation framework itself is not measuring what it claims.

### 7.3. For Safety and Alignment Teams

First, anticipate unintended consequences. Interventions based on flawed diagnostics may degrade performance for populations whose cognitive patterns differ from benchmark norms. Before "correcting" apparent failures, ask whether the behavior you are modifying is actually problematic or simply different from one cultural group's expectations.

Second, monitor for alignment tax across diverse populations. If alignment procedures improve performance on benchmarks derived from Western participants while degrading performance for other groups, you have not solved a problem. You have inscribed a bias.

Third, invest in understanding before intervening. The entanglement of LLM representations means that targeted modifications produce unpredictable broad effects. We don't fully understand these systems yet and we need to act like it.

### 7.4. For Reviewers and Program Committees

First, require cultural specification in evaluation claims. Papers asserting "human-like" performance should be expected to clarify which humans, from which contexts, using which norms. Treat unqualified universality claims as methodological weakness.

Second, value evaluation critique. Position papers and empirical work that question dominant evaluation frameworks serve the field even when they do not propose immediate replacements. Creating space for this work is essential if the community is to course-correct.

Third, reward engagement with cognitive science literature. The benchmarks in question originate in cognitive science, where debates about universality and cultural variation have unfolded for decades. ML researchers borrowing these tools should engage with that literature and reviewers should expect them to.

## 8. Limitations

Our argument focuses on cognitive benchmarks derived from human psychological research. We do not claim that all LLM evaluation is flawed. Benchmarks measuring factual accuracy, coding ability, or task completion are outside our scope.

We also do not offer a fully developed alternative evaluation framework. We argue that current approaches are broken, but building replacements is work that remains to be done.

Finally, our critique draws primarily on cross-cultural cognitive research, which itself has limitations, often relying on binary East/West comparisons that oversimplify human diversity. A stronger foundation would draw on more granular cultural research than currently exists.

## 9. Conclusion

The machine learning community has adopted human cognitive benchmarks to evaluate whether LLMs develop "human-like" concept representations. This paper has argued that this approach rests on a chain of unsupported assumptions—a house of cards—each one necessary for the process to make sense, and each one questionable.

The benchmarks assume that LLM outputs reflect stable internal structures. They do not. Minor changes in prompting, formatting, and option ordering produce dramatic shifts in results, providing evidence that these evaluations measure context-dependent responses rather than durable representations.

The benchmarks assume that human cognition provides a universal standard. It does not. Fifty years of cross-cultural research documents systematic variation in how people categorize, reason, and perceive. There is no single "human" way of organizing concepts against which LLMs can be measured.

When LLM outputs deviate from benchmark norms, researchers may interpret this as failure. But that interpretation isn't built into the benchmarks, it's a choice. And when studies using the same methods reach incompatible conclusions (LLMs align with human cognition, LLMs fall short, LLMs exceed human performance), maybe the problem isn't the models. Maybe it's the assumption that deviation means something went wrong.

The same researchers interpreting deviation as failure may also believe that diagnosed failures should be corrected. But interventions based on flawed diagnostics risk real harm: degraded reasoning capabilities, forgotten pre-training abilities, and alignment procedures that inscribe the preferences of dominant populations while penalizing diversity.

And there you have our house of cards.

We are not arguing that LLM evaluation is impossible or that these systems are beyond scrutiny. We are arguing that the current paradigm of treating culturally specific cognitive benchmarks as universal standards, frequently interpreting deviation as deficiency, and intervening to correct it, isn't helpful to progressing our understanding or LLMs or making them better aligned and safer. LLMs are statistical compression systems trained on global, multilingual data, and our evaluations of them need to keep that in mind.

The field needs evaluation approaches designed for what LLMs actually are. Until we develop them, humility is warranted when it comes to what benchmarks measure, what "failure" means, and the wisdom of interventions built on foundations this uncertain.

## Generative AI Statement

This paper was drafted with assistance from Claude (Anthropic) followed by heavy revisions by the author. The core arguments, concepts, analysis, and position are the author's own. Claude was also used for editing support and as a sounding board.

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
