# OpenReview forum: "Position: Stop Using Culturally Biased Human Cognitive Benchmarks to Evaluate LLMs"
_ICML.cc/2026/Position_Paper_Track — ICML 2026 Position Paper Track regular_

### Official Review · Reviewer_vPH7 · 2026-03-12

**Significance:** 4
**Argument Clarity:** 3
**Rating:** 4
**Confidence:** 3

**Questions:**

See weaknesses

Ryan et al. (2010), -> Ryan et al. (2024)?

**Alternative Views Section:**

Yes

**Compliance With Llm Reviewing Policy A Conservative:**

Affirmed.

**Discussion Potential:**

3

**Paper Summary:**

The paper argues that current human cognitive benchmarks that are being used as a proxy to measure human-like understanding are misguiding, not representative of human cognition and can lead to interventions that compromise rather than help improve LLMs (e.g. losing diverse pre-training knowledge at the expense of learning human preference collected from a narrow human population). They use a house of cards analogy to using such benchmarks that helps understand why this is fundamentally flawed - from misrepresentation of LLM internal state which change with minor changes to context to overfitting to cognitive behavior of specific groups when claiming human cognition to using them as a source of truth for designing flawed interventions.

**Position:**

Yes

**Position In Title:**

Yes

**Related Work:**

2

**Strengths And Weaknesses:**

Strengths:
- the claims are well crafted and supported and the house of cards framework makes the argument very cohesive
- the call to action gives concrete pragmatic steps on benchmarks, results interpretation, definition of success

Weaknesses
- the argument that LLM pre-training knowledge is global does not hold in all settings as matching language specific LLMs to local preferences might not be flawed
- while true that LLMs are sensitive to prompt changes and that model outputs may change as a result, however, it could be just how we measure "the internal state" that is flawed.

**Support:**

3

---

> ### Author Rebuttal · Authors · 2026-03-31
>
> Thank you for your review and careful reading that caught our citation error, which we will correct. We are pleased the house of cards metaphor resonated.
>
> Regarding the language-specific LLM point, we appreciate the nuance and agree that, in theory, it might be possible for a language-specific LLM to reflect local practices. Nevertheless, we think that there are too many variables that would have to be satisfied to make this possible in reality. Even apparently homogeneous linguistic groups contain meaningful cognitive diversity across qualities like age, gender, class, education, geographic location. Furthermore, a qualifying language would have to be free of any colonial history, dialect variation, or generational drift. Additionally, the corpora and the benchmarked population would have to perfectly align temporally. It’s hard to imagine a language let alone a population that could meet these conditions.
>
> Your point about prompt sensitivity and inadequate measurement of internal states is well taken. However, ultimately, whether internal states are genuinely unstable or simply inaccessible to current measurement tools, the limitations of the benchmarks remain unchanged. Human cognitive benchmarks cannot reliably be used to measure what researchers claim to measure. Therefore, even if future interpretability research demonstrates stable internal states exist, Cards 2, 3, and 4 are sufficient to topple the house.

---

### Official Review · Reviewer_XmnJ · 2026-03-13

**Significance:** 3
**Argument Clarity:** 3
**Rating:** 6
**Confidence:** 3

**Questions:**

I have already raised several questions regarding this paper. Please find them in the Weakness section above.

P.S. My understanding about the objectives of position papers is that they are always more about pointing out the issues, rather than offering concrete solutions. But the issues should preferably be well articulated, nevertheless. For this very reason, I’m very much looking forward to the authors’ response.

**Alternative Views Section:**

Yes

**Compliance With Llm Reviewing Policy A Conservative:**

Affirmed.

**Discussion Potential:**

4

**Final Justification:**

I believe the authors have managed to fully resolve my original concerns in the rebuttal process. Thanks to all of these, I'm raising the rating to 6 from 5.

**Paper Summary:**

This paper boils down to one central demand: **stop using human cognitive benchmarks, which are sampled from a limited population source, as universal standards for evaluating LLMs’ understanding of concepts**. The authors argue that such practice fails to realize that the actual human cognition can be swayed by a number of specific factors, including but not limited to personal preferences or cultural upbringings. The current usage of human cognitive benchmarks for LLMs often ignores such specificity, instead assuming humans are always stable and share a universal Western culture background. As evidence, the authors show multiple studies, which are built on such biased benchmarking practice, return contradictory and flawed findings, as a result.

**Position:**

Yes

**Position In Title:**

Yes

**Related Work:**

3

**Strengths And Weaknesses:**

Since reviewing such 'position' papers is new to me, rather than talking in terms of the conventional *strengths vs. weaknesses*, I find it more comfortable to express my opinions as **Hits and Misses** based on my own perspectives and expertise, regarding the authors' proposed positions.

For the record, my background comes from **fairness in designing evaluative benchmarks**.  I also have experience in **image-text alignment**.

---

### **The Hits, aka "Strengths"**

- The authors have done a decent job showing convincing evidences for the symptoms. In particular, my favorite is the spoon-fork-chopsticks example in Card 2 - it is a simple yet brilliantly perceptive example that demonstrates of the culturally loaded elements hidden under many cognitive LLM benchmarks that are marketed as universal.
- The authors’ point about contradictory findings is also well-taken. When the same type of test produces wildly different conclusions depending on who runs it and how, such as by the many studies using the THINGS database, it leaves me convinced that it is usually the evaluative paradigm/pipeline itself is to blame.

### **The Misses, aka "Weaknesses" or Concerns**

- **The implicit goal of culturally universal evaluation may be practically infeasible.** The authors convincingly demonstrate that existing human cognitive benchmarks reflect narrow cultural contexts. But I find their proposed remedy stated in Section 7.1 - developing "culturally pluralistic benchmarks" that "treat variation as information rather than noise" - may face a feasibility problem. Cultural cognition exists on **a continuum shaped by intersecting factors** like language, geography, religion, education, and individual experience; even within a single country, such as those as vast as USA or China, cognitive patterns can already vary dramatically on a town or community basis. This begs my questions regarding the feasibility of this goal -
  * How many distinct cultural reference points are enough to considered ‘culturally pluralistic’?
  * And at what point does the benchmark become so *expansive* that it ceases to function as a practical tool?

**Support:**

4

---

> ### Author Rebuttal · Authors · 2026-03-31
>
> Thank you for your positive assessment of our work. We are in agreement that culturally universal evaluation is practically infeasible. We would go further and argue that it is also undesirable. In recommending culturally pluralistic benchmarks, we explicitly clarify in Section 7.1, “This does not mean adding a few non-Western participants to existing frameworks. It means designing evaluation approaches that treat variation as information rather than noise such as benchmarks that characterize how model outputs relate to different cultural contexts rather than measuring deviation from a single standard.”
>
> We concur that intracultural cognitive diversity produces meaningful cognitive variation across the factors you identify and develop this point further in our response to X5rz. Given that no benchmark can achieve comprehensive cultural representation, Section 7.1 also asks whether the goal should be consistency with human cognition at all, suggesting that evaluation strategies that characterize emergent patterns may be more useful than those that measure alignment with a particular population. Moreover, honest acknowledgment of what a benchmark measures and for whom is both more feasible and more epistemically sound than any attempt at universality. The questions of how many reference points are sufficient and when a benchmark becomes too expansive assume a comprehensiveness goal that our recommendation explicitly rejects.

---

> > ### Author Rebuttal · Reviewer_XmnJ · 2026-04-03
> >
> > I'd like to see if the authors could further elaborate on what they mean by 'emergent patterns' with at least one concrete example, if the time permits.
> >
> > With all being said, I believe we are both clear that 'a culturally comprehensive benchmark is infeasible to obtain' - this core statement should have been more explicitly stated early on in the paper (as early as Card 2) for better clarity.

---

### Official Review · Reviewer_X5rz · 2026-03-13

**Significance:** 2
**Argument Clarity:** 2
**Rating:** 3
**Confidence:** 4

**Questions:**

## Key Questions

- If the community fully adopts your call to abandon human cognitive benchmarks, what specific novel metrics should be used to objectively measure an LLM's "concept understanding" and "internal representations"?

- Prompt sensitivity might be resolved as LLM architectures evolve; if future models achieve high prompt robustness and your first "card" falls, does your overall stance against cognitive benchmarks remain intact ?

- How do you reconcile the claim that LLMs lack stable internal conceptual structures with your citation of Betley et al., which shows that targeted fine-tuning can reliably shift behaviors across contexts ?

- When rebutting Alternative 4, you note that averages skew toward dominant cultures; does this imply that if we could build a perfectly balanced global training corpus, the existing evaluation logic would become valid again?

- You advise alignment teams to "monitor for alignment tax across diverse populations," but without the standardized benchmarks you heavily critique, how exactly should engineers quantitatively conduct this monitoring?

- Your citations of cross-cultural psychology focus primarily on visual categorization and concept association; does this "cultural difference bias" apply equally to more foundational cognitive benchmarks like logical reasoning, mathematics, or spatial cognition?

- For massive, historically established datasets like THINGS, how do you practically recommend that evaluation designers retroactively document the cultural context of their norms?

- If we entirely abandon human norms as a baseline, how can we differentiate between an emergent pattern born from multicultural global data and a genuine logical failure in an LLM's output?

- Given that you acknowledge current cross-cultural studies often rely on oversimplified binary comparisons, does relying heavily on these studies inadvertently reinforce the flawed measurement systems you are critiquing?

- If LLMs are fundamentally engines of statistical compression over global data, is demanding that their evaluation frameworks reflect true human diversity an act of anthropomorphism toward a purely probabilistic model?

**Alternative Views Section:**

Yes

**Compliance With Llm Reviewing Policy A Conservative:**

Affirmed.

**Discussion Potential:**

2

**Ethics Review Area:**

["Other Expertise"]

**Final Justification:**

Thanks for your detailed response. Overall, the current recommendation is appropriate.

**Paper Summary:**

This paper argues against evaluating LLMs using human cognitive benchmarks. These tests falsely assume models have stable internal states and mistakenly treat culturally narrow Western norms as universal standards. Relying on them yields contradictory research and drives harmful alignment interventions. The authors call for novel evaluation methods tailored to LLMs as global statistical systems rather than human approximations.

**Position:**

Yes

**Position In Title:**

Yes

**Related Work:**

3

**Strengths And Weaknesses:**

## Strengths

- The core argument accurately targets a critical pain point in LLM evaluation regarding cultural bias and flawed cognitive measurement.
- The cross-disciplinary integration of cross-cultural cognitive psychology, such as holistic versus analytic cognition, effectively supports the claim that no universal human cognitive standard exists.

- Linking flawed benchmark diagnostics to real-world harms like the "safety tax" and bias perpetuation significantly elevates the paper's practical urgency and policy relevance.

## Weaknesses

- By explicitly admitting the lack of an alternative evaluation framework, the paper focuses heavily on deconstruction without offering constructive solutions, which limits its practical utility.
- The reliance on traditional binary psychological studies, such as East versus West comparisons, oversimplifies human diversity, a limitation the authors acknowledge but fail to fully mitigate.
- Presenting prompt sensitivity and cultural bias as parallel core arguments feels disjointed, as the former is an architectural robustness issue while the latter concerns representation logic.
- The frequent use of informal colloquialisms, such as "killing a bunch of birds", "drop of a hat", and "umpteen times", undermines the professional and restrained tone expected at a top-tier conference like ICML.

**Support:**

2

---

> ### Author Rebuttal · Authors · 2026-03-31
>
> Thank you for your thoughtful review. As per the ICML Position Paper CFP, the goal of the position track is to highlight papers that stimulate constructive, civil discussion on timely topics that need our community's attention. That is the practical utility of this paper as your engagement has so clearly demonstrated.
>
> We agree that LLMs are fundamentally engines of statistical compression over global data which is why we do not recommend that their evaluation frameworks reflect true human diversity. It follows that human norms are not the appropriate baseline for distinguishing emergent patterns from genuine failures. Logical failures should be evaluated on logical grounds using benchmarks designed around what we actually know about LLM behavior rather than what we expect from human cognition. As we state in the conclusion, “The field needs evaluation approaches designed for what LLMs actually are." While specifying novel metrics and alternative evaluation frameworks falls outside the scope of a position paper, we have included a call to action targeting multiple stakeholders.
>
> Our primary request of evaluation designers is that they are honest and transparent about which populations the benchmarks measured and consequent limitations linked to that specificity. We ask that benchmarks not be used to generalize to all humans. We also are not opposed to the use of standardized benchmarks by alignment teams. We are opposed to the application of benchmarks that are inherently flawed for multiple reasons as is the case with human cognitive benchmarks.
>
> The entire point of our paper is that human cognitive benchmarks oversimplify human diversity. We acknowledge your specific concern about binary psychological studies and indicate the need for more granular cultural research than currently exists. At the same time, greater diversity would only support our thesis. To our knowledge, the more granular cross-cultural cognitive research that would be needed to address the concern about mitigating the limitations of binary comparisons does not yet exist at scale, which is why we flag it in our Limitations section. Note that we cite cross-cultural research for a fundamentally different purpose than the benchmarks we critique. We use it as existence proof that cognitive variation exists (a purpose for which even imperfect binary comparisons are sufficient) rather than as a normative standard against which to measure correct cognition. If you are aware of relevant citations we should have included, that would be helpful for the camera ready version.
>
> The use of the house of cards metaphor was deliberate because if even one card falls, those dependent on it also fall. The purpose of citing Betley et al. was to demonstrate the trouble with Card 1 and the lack of stable internal conceptual structures. Yes, targeted fine-tuning reliably shifts behaviors, however, it does so in bizarre and unpredictable ways. But even if prompt sensitivity were to be resolved, Card 2 on the lack of universality would take down the rest. Therefore, our overall stance would remain intact.
>
> We present prompt sensitivity and cultural bias as parallel core arguments because, while they are mechanistically distinct as you point out (the former is an architectural robustness issue and the latter concerns representation logic), both undermine the same foundational assumption that human cognitive benchmark results reflect something stable and meaningful about an LLM's cognition. We attempt to cover all lines of potential skepticism in the hopes of reaching readers whose doubts about our position might stem from various areas of concern. Please see our response to vPH7 on language-specific models for our thoughts on the feasibility of a perfect training corpus.
>
> Regarding cross cultural bias and more foundational cognitive benchmarks, yes, we believe the concern extends to those domains though the specific mechanisms may differ. Logical reasoning benchmarks embed assumptions about valid argument structure, formalism, and what counts as sufficient evidence that are not culturally neutral. Spatial cognition varies systematically. For example, absolute vs relative directional reasoning is well documented. Research in mathematics education underscores that there is no singular correct cognitive path from problem to solution. We would welcome engagement with specific benchmarks where researchers believe cultural bias may or may not apply as this is precisely the kind of discussion position papers are designed to stimulate.
>
> Given the explicit goals of a position paper, we wanted the paper to be accessible to as many readers as possible. We acknowledge the feedback on tone and would revisit the flagged expressions for the camera-ready version.

---

> > ### Author Rebuttal · Reviewer_X5rz · 2026-04-04
> >
> > The author has resolved some of my issues. I will retain the original score.

---

### Official Review · Reviewer_na4Q · 2026-03-16

**Significance:** 2
**Argument Clarity:** 2
**Rating:** 2
**Confidence:** 3

**Questions:**

What is human-cognitive benchmarks, any definition, related work and examples?

How significant is the use of those datasets?

How to measure biases and impacts those datasets caused?

How to improve and mitigate the use of human cognitive benchmark realistically?

**Alternative Views Section:**

Yes

**Compliance With Llm Reviewing Policy A Conservative:**

Affirmed.

**Discussion Potential:**

2

**Paper Summary:**

This paper advocates the stop of using based evaluation datasets, particularly, 'human-cognitive benchmarks'. Those benchmarks are generated by human from particular groups and may introduce unintended biases. The authors argues that using cognitive benchmarks can lead to misguided interventions. The authors provided several viewpoints on why this can lead to incorrect and biased generation.

**Position:**

Yes

**Position In Title:**

Yes

**Related Work:**

1

**Strengths And Weaknesses:**

Strengths:
The paper studies and advocates awareness of biases in LLM benchmarks.

Weaknesses:
The paper didn't clearly explain in depth on the impact of biased datasets. For example, what are the popular such benchmarks and what are their impact. How to measure such impact and bias caused by evaluating using those benchmarks?

The paper also didn't provide clear definition of human cognitive benchmark. It might also be good to discuss different aspects of cognitive benchmark and their effects in different stages of training LLM, e.g., pre-training, fine-tuning, RL, etc.

The paper didn't provide alternate approaches of not using such benchmarks. For example, how to generate unbiased datasets. How to measure bias in existing datasets. How to correct model tuned based on those datasets. Simply not using a possibly large portion of evaluation benchmarks without providing a solution limits the potential of this paper.

**Support:**

2

---

> ### Author Rebuttal · Authors · 2026-03-31
>
> Thank you for your review. We wish to clarify that our focus is human cognitive benchmarks and the problems associated with their application to LLMs, rather than bias in datasets or training data composition. The introduction describes these benchmarks as "prepackaged, ready-to-administer categorization tasks, similarity judgments, and concept organization assessments that come with preestablished human norms." Specific examples include Rosch's typicality norms (derived from American undergraduates in the 1970s) and the THINGS database, both discussed in Section 4. We acknowledge that formalizing this definition further would aid readers less familiar with the cognitive science literature and would do so for the camera-ready version.
>
> Section 4 also documents the concrete impact of applying these benchmarks to LLMs: studies using these identical tools reach directly contradictory conclusions (Du et al., 2025; Studdiford et al., 2025; Hrytsyna & Alves, 2025; Shani et al., 2025), which we argue is itself evidence that the framework is broken. Although we reference training data at times, it is always in service of our argument about evaluation methodology. Our related work deliberately spans cognitive science, ML, and alignment research, as the argument is inherently cross-disciplinary.
>
> While questions about measuring bias and discussing training stages fall outside the scope of this paper as described above, we address the training data question directly in Section 6.3. On the question of alternatives, we would note that clearly identifying a broken framework is itself a contribution, one the field needs before productive replacement work can begin. That said, Section 7 offers concrete directional recommendations for evaluation designers, researchers, alignment teams, and reviewers. A complete alternative framework is beyond the scope of a position paper, and we state this openly in our Limitations section.

---

> > ### Author Rebuttal · Reviewer_na4Q · 2026-04-05
> >
> > thank you for your response! I will keep my rating.

---

### Decision · Program_Chairs · 2026-04-30

**Decision:**

Accept (regular)

**Comment:**

This paper points out that many benchmarks are culturally biased or misinformed.  Human-like understanding is not universal in the way it is discussed in some LLM work.  They use the metaphor of a house of cards to illustrate negative impacts of this assumption.

Reviewers agreed with this position, but some reviewers wanted to know what should be done in response to this problem.  The authors state “specifying novel metrics and alternative evaluation frameworks falls outside the scope of a position paper, we have included a call to action targeting multiple stakeholders.”

In general I side with the authors here, this is a paper that raises many questions without a lot of solutions.  But I would claim that’s ok for the position paper track.  It is worthwhile to raise concerns before solutions are found.